# UNDERSTANDING UNFAIRNESS VIA TRAINING CONCEPT INFLUENCE

## ABSTRACT

Identifying the causes of a model's unfairness is an important yet relatively unexplored task. We look into this problem through the lens of training data – the major source of unfairness. We ask the following questions: how would the unfairness of a model change if its training samples (1) were collected from a different (*e.g.* demographic) group, (2) were labeled differently, or (3) whose features were modified? In other words, we quantify the influence of training samples on unfairness by counterfactually changing samples based on predefined concepts, *i.e.* data attributes such as features ($X$), labels ($Y$), and sensitive attributes ($A$). To compute a training sample's influence on the model's unfairness w.r.t a concept, we generate *counterfactual samples* based on the concept, *i.e.* the counterfactual versions of the sample if the concept were changed. We then calculate the resulting impact on the unfairness, via *influence function* (Koh & Liang, 2017; Rousseeuw et al., 2011), as if the counterfactual samples were used in training. Our framework not only can help practitioners understand the observed unfairness and mitigate it by repairing their training data, but also leads to many other applications, *e.g.* detecting mislabeling, fixing imbalanced representations, and detecting fairness-targeted poisoning attacks.

## 1 INTRODUCTION

A fundamental question in machine learning fairness is: what causes unfairness? Without knowing the answer, it is hard to understand and fix the problem. In practice, this is also one of the first questions the practitioners would ask after computing the fairness measures and finding the model to be unfair. Although the question sounds simple, it is hard to identify the exact source of unfairness in the machine learning pipeline, as admitted by many leading fairness practitioners, *e.g.* Meta blog (Bogen & Corbett-Davies, 2021) states: "Unfairness in an AI model could have many possible causes, including not enough training data, a lack of features, a misspecified target of prediction, or a measurement error in the input features. Even for the most sophisticated AI researchers and engineers, these problems are not straightforward to fix."

The sources of unfairness are many, including data sampling bias or under-representation (Chai & Wang, 2022; Zhu et al., 2022; Celis et al., 2021; Bagdasaryan et al., 2019), data labeling bias (Wang et al., 2021; Wu et al., 2022b; Fogliato et al., 2020), model architecture (or feature representation) (Adel et al., 2019; Madras et al., 2018; Zemel et al., 2013; Song et al., 2019; Xing et al., 2021; Li et al., 2021a; Song et al., 2021; Li et al., 2020), distribution shift (Ding et al., 2021; Chen et al., 2022; Rezaei et al., 2021; Giguere et al., 2022) *etc.* In this work, we tackle this problem by looking at the most important and obvious source of bias: the training samples. If the model's training samples are biased, then it would be highly challenging for the model to remain fair. Specifically, we ask the following questions regarding how training samples would impact the model's unfairness: how a model's (un)fairness would change if its training samples (1) were collected from a different (*e.g.* demographic) group, (2) were labeled differently, or (3) were modified for some features? Answering those questions can help practitioners (1) *explain* the cause of the model's unfairness in terms of training data, (2) *repair* the training data to improve fairness, and (3) *detect* biased or noisy training labels, under-represented group, and corrupted features that hurt fairness.

In this work, we measure the training sample's impact on fairness using *influence function* (Cook & Weisberg, 1982; Koh & Liang, 2017), and we define the influence on fairness measure w.r.t a

training *concept*, *i.e.* a categorical variable that describes data property. For example, we can choose the concept to be the sensitive group attribute and counterfactually[1] change it to answer the question "What is the impact on fairness if the training data were sampled from a different group?" Or we can choose the concept to be the training labels, and then our method measures the impact on fairness when the label is changed. We can also apply the concept to the training features. Our flexible framework generalizes the prior works that only consider removing or reweighing training samples (Wang et al., 2022a; Li & Liu, 2022), and we can provide a broader set of explanations and give more insights to practitioners in a wider scope (*e.g.* what if a data pattern is drawn from another demographic group?). We name our influence framework as *Concept Influence for Fairness* (CIF).

In addition to explaining the unfairness, CIF can also recommend practitioners ways to fix the training data to improve fairness by counterfactually changing concepts in training data. Furthermore, we demonstrate the power of our framework in a number of other applications including (1) detecting mislabeling, (2) detecting poisoning attacks, and (3) fixing imbalanced representation. Through experiments on 4 datasets – including synthetic, tabular, and image – we show that our method achieves satisfactory performance in a wide range of tasks.

## 2 INFLUENCE OF TRAINING CONCEPTS

We start with introducing the influence function for fairness, the concept in training data, and define our *Concept Influence for Fairness* (CIF).

### 2.1 FAIRNESS INFLUENCE FUNCTION

**Influence Function on Group Fairness.** Denote the training data by $D_{train} = \{z_i^{tr} = (x_i^{tr}, y_i^{tr})\}_{i=1}^n$ and the validation data by $D_{val} = \{z_i^{val} = (x_i^{val}, y_i^{val})\}_{i=1}^n$. Suppose the model is parameterized by $\theta \in \Theta$, and there exists a subset of training data with sample indices $\mathcal{K} = \{K_1, ..., K_k\}$. If we perturb a group $\mathcal{K}$ by assigning each sample $i \in \mathcal{K}$ with weight $w_i \in [0, 1]$, denote the resulting counterfactual model's weights by $\hat{\theta}_{\mathcal{K}}$.

**Definition 1.** *The fairness influence of reweighing group $\mathcal{K}$ in the training data is defined as the difference of fairness measure between the original model $\hat{\theta}$ (trained on the full training data) and the counterfactual model $\hat{\theta}_{\mathcal{K}}$:*

$$\text{infl}(D_{val}, \mathcal{K}, \hat{\theta}) := \ell_{fair}(\hat{\theta}) - \ell_{fair}(\hat{\theta}_{\mathcal{K}}) \tag{1}$$

where $\ell_{fair}$ is the fairness measure (will be specified shortly after).

Similar to (Koh & Liang, 2017; Koh et al., 2019; Li & Liu, 2022), we can derive the closed-form solution of fairness influence function (see Appendix A for the derivation):

**Proposition 1.** *The first-order approximation of $\text{infl}(D_{val}, \mathcal{K}, \hat{\theta})$ takes the following form:*

$$\text{infl}(D_{val}, \mathcal{K}, \hat{\theta}) \approx -\nabla_\theta \ell_{fair}(\hat{\theta})^\intercal H_{\hat{\theta}}^{-1} \left( \sum_{i \in \mathcal{K}} w_i \nabla \ell(z_i^{tr}; \hat{\theta}) \right) \tag{2}$$

*where $H_{\hat{\theta}}$ is the hessian matrix i.e. $H_{\hat{\theta}} := \frac{1}{n} \nabla^2 \sum_{i=1}^n \ell(z_i^{tr}; \hat{\theta})$, and $\ell$ is the original loss function (e.g. cross-entropy loss in classification).*

**Approximated Fairness Loss.** The loss $\ell_{fair}(\hat{\theta})$ quantifies the fairness of a trained model $\hat{\theta}$. Similarly to prior work (Wang et al., 2022a; Sattigeri et al., 2022), we can approximate it with a surrogate loss on the validation data. Denote the corresponding classifier for $\theta$ as $h_\theta$, we can approximate the widely used group fairness Demographic Parity (Calders et al., 2009; Chouldechova, 2017) (DP)

---

[1] We use the word "counterfactual" in its literal sense, *i.e.* being different from the factual world, in the same *empirical* fashion of the counterfactual example or the counterfactual explanation (Verma et al., 2020; Ustun et al., 2019; Roese, 1997) rather than in the rigorous and theoretical sense of "counterfactual" in the causal inference. Our work does *not* belong to the area of causal inference.

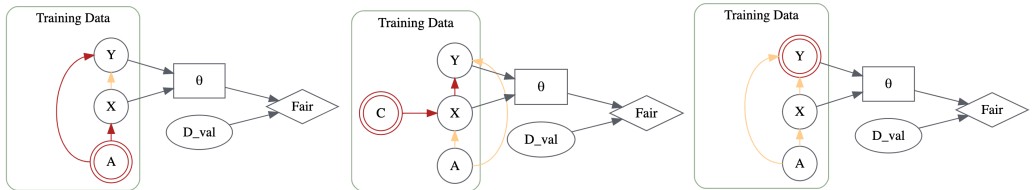

(a) Overriding sensitive attribute $A$. (b) Overriding feature $X$. (c) Overriding label $Y$.

Figure 1: Our data dependency assumption. Yellow arrows represent the data dependency link and red arrows represent the effect of overriding (*i.e.* counterfactually changing the value of a concept). In training data, the concept variable $C$ can override sensitive attribute $A$ (i.e. Figure (a)), features $X$ (i.e. Figure (b)), and label $Y$ (i.e. Figure (c)). We train the model $\theta$ on $X$ and $Y$, and compute the validation fairness metric Fair on the validation dataset $D_{val}$.

violation as the following (assume both $A$ and the classification task are binary):

$$\ell_{DP}(\hat{\theta}) := \left| \mathbb{P}(h_\theta(X) = 1 | A = 0) - \mathbb{P}(h_\theta(X) = 1 | A = 1) \right| \tag{3}$$

$$\approx \left| \frac{\sum_{i \in D_{val}:a_i=0} g(z_i^{val};\theta)}{\sum_{i \in D_{val}} \mathbb{I}[a_i = 0]} - \frac{\sum_{i \in D_{val}:a_i=1} g(z_i^{val};\theta)}{\sum_{i \in D_{val}} \mathbb{I}[a_i = 1]} \right| \tag{4}$$

where $g$ is the logit of the predicted probability for class 1. See Appendix C for the approximated violation of Equality of Opportunity (Hardt et al., 2016) (EOP) and Equality of Odds (Woodworth et al., 2017) (EO).

## 2.2 CONCEPTS IN TRAINING DATA

A concept is a *sample-level* categorical attribute associated with the training data. Formally, denote a concept by $C \in \mathcal{C} := \{1, 2, ..., c\}$ where $C$ is a discrete concept encoded in the dataset $(X, Y, A)$. $C$ can simply be either $Y$ or $A$ or any feature in $X$ or in a broader definition. See Figure 1 for an illustration of the training concepts and our data dependency assumption. Our core idea is to quantify the influence when each training sample is replaced by its *counterfactual sample*, *i.e.* the counterfactual version of the sample if its concept were changed, when we transform the sample w.r.t. a certain concept. We use the term *override* to mean counterfactually changing the concept, *e.g.* overriding a sample's concept to $c$ means counterfactually setting its concept to $c$ and replacing the sample with its counterfactual version as if its concept were $c$. We will formally define overriding in Section 2.3.

**Examples.** We provide examples of concepts and motivate why transforming samples based on training concepts can be intuitively helpful for fairness.

- **Concept as Sensitive Attribute** ($C = A$). Intuitively speaking, the sensitive/group attribute relates closely to fairness measures due to its importance in controlling the sampled distribution of each group. Changing $A$ corresponds to asking counterfactually what if a similar or counterfactual sample were from a different sensitive group.

- **Concept as Label** ($C = Y$). In many situations, there are uncertainties in the label $Y|X$. Some other times, the observed $Y$ can either encode noise, mislabeling or subjective biases. They can all contribute to unfairness. Changing $Y$ implies the counterfactual effect if we were to change the label (*e.g.* a sampling, a historical decision, or a human judgment) of a sample.

- **Concept as Predefined Feature Attribute** ($C = attr(X)$). Our framework allows us to predefine a relevant concept based on feature $X$. $C$ can be either an externally labeled concept (*e.g.* sample-level label in image data) or a part of $X$ (*e.g.* a categorical[2] feature in tabular data). For instance, if we want to understand how skin color would affect the model's fairness, and if so which data samples would impact the observed fairness the most w.r.t skin color, we can specify

---

[2]All concepts in $X$, $Y$, or $A$ that we consider are assumed to be categorical because the continuous concept is not well-defined in the literature of concept.

$C = attr(image) \in \{\text{dark}, \text{light}\}$. Then transforming w.r.t. this concept corresponds to identifying samples from different skin colors that, if were included in the training data, would lead to a fairer model.

- **Concept as Removal.** Our setting is also flexible enough to consider the effect of removing a training sample, as commonly considered in the literature on influence function (Li & Liu, 2022). Consider a selection variable $S \in \{1, 0\}$ for each instance $z_i^{tr}$, for each sample that appears in the training data we have $s_i = 1$. Changing to $s_i = 0$ means the sample is counterfactually removed, *i.e.* $\hat{z}_i^{tr}(c') = \varnothing$. By allowing the removal concept, we can incorporate the prior works on the influence of removing samples into our framework.

## 2.3 CONCEPT INFLUENCE FOR FAIRNESS (CIF)

Our goal is to quantify the counterfactual effect of changing concept $c$ for each data sample $(x, y, a)$. Mathematically, denote by $(\hat{x}, \hat{y}, \hat{a})$ the counterfactual sample by overriding $c$. Consider a training sample $z_i^{tr} := (x_i, y_i, a_i, c_i)$, and define a counterfactual sample for $z_i^{tr}$ when counterfactually changing from $C = c$ to $C = c'$ as follows:

$$\hat{x}(c'), \hat{y}(c'), \hat{a}(c') = tranform(X = x, Y = y, A = a, override(C = c')), \ c' \neq c. \tag{5}$$

In the definition above, $override(\cdot)$ operator counterfactually sets the value of the concept variable to a different $c'$. If differs from merely $C = c$ in the sense that the change on $C$ would also change other variables, *i.e.* $A$, $X$, $Y$, or the action of removing samples. It also differs from the well-known *do*-operator in the causal literature (Pearl, 2010), in the sense that the procedure does not necessarily need to follow the mechanism of causal inference, and therefore can include any *empirical* mechanisms that approximate counterfactuals. The difference is vital because when it is unclear whether the problem is identifiable or not (Zhang & Hyvärinen, 2009; Zhang & Hyvarinen, 2012; Shimizu et al., 2006; Hoyer et al., 2008), we still want to develop heuristic approximations. And $transform(\cdot)$ maps the original training samples to their corresponding counterfactual samples by considering the effect of $override(\cdot)$.

Therefore, our $override(\cdot)$ and $transform(\cdot)$ can include both traditional causal inference methods when we deal with synthetic data, and, more importantly, empirical heuristics when identifiability is unclear. We include three typical scenarios:

- Assigning values: When we override label $Y$, we can simply set the label value (which is not a typical case in causal inference), *i.e.*

$$tranform(X = x, Y = y, A = a, override(Y = y')) = (x, y', a)$$

- Empirical approximation: When identifiability is unclear, *which is the major case that we study*, we can approximate the counterfactual samples by training a generative model $G$,[3] *i.e.*

$$transform(X = x, Y = y, A = a, override(C = c')) = G_{c \to c'}(x, y, a)$$

- *Do*-intervention: When the counterfactual distribution is theoretically identifiable, which is only in the synthetic setting, then the $transform(\cdot)$ and $override(\cdot)$ are the sampling functions and the *do*-operator, *i.e.* [4]

$$tranform(X = x, Y = y, A = a, override(C = c')) \tag{6}$$

$$= \hat{x}(c'), \hat{y}(c'), \hat{a}(c') \sim \mathbb{P}\left(\hat{X}, \hat{Y}, \hat{A} | X = x, Y = y, A = a, do(C = c')\right), \ c' \neq c. \tag{7}$$

The effectiveness of our solution depends on finding a proper $transform(\cdot)$, which is our work's focus, and the quality of $transform(\cdot)$ can be verified empirically in experiments. In addition, our framework is general; if researchers discover better ways to approximate counterfactuals, they plug those into our framework easily.

**Concept Influence for Fairness.** Denote a counterfactual sample as $\hat{z}_i^{tr}(c') = (\hat{x}_i(c'), \hat{y}_i(c'), \hat{a}_i(c'), \hat{c}_i = c')$. Then we define the counterfactual model when replacing $z_i^{tr} = (x_i, y_i, a_i, c_i)$ with $\hat{z}_i^{tr}(c')$ as:

$$\hat{\theta}_{i,c'} := \operatorname{argmin}_\theta \{R(\theta) - \epsilon \cdot \ell(\theta, z_i^{tr}) + \epsilon \cdot \ell(\theta, \hat{z}_i^{tr}(c'))\} \tag{8}$$

---

[3]In Section 3.1, we introduce how to construct $transform(\cdot)$ in this case.

[4]The definition is slightly abused – when $C$ overlaps with any of $(X, Y, A)$, the $do(\cdot)$ operation has a higher priority and is assumed to automatically overwrite the other dependencies. For example, when $C = A$, we have: $\mathbb{P}\left(\hat{X}, \hat{Y}, \hat{A} | X = x, Y = y, A = a, do(C = c')\right) = \mathbb{P}\left(\hat{X}, \hat{Y}, \hat{A} | X = x, Y = y, \cancel{A = a}, do(A = \hat{a})\right)$.

**Definition 2** (Concept Influence for Fairness (CIF)). *The concept influence for fairness (CIF) of overriding on a concept $C$ to $c'$ in sample $i$ on the fairness loss $\ell_{fair}$ is defined as:*

$$infl(D_{val}, \hat{\theta}_{i,c'}) := \ell_{fair}(\hat{\theta}) - \ell_{fair}(\hat{\theta}_{i,c'}) \tag{9}$$

Based on Proposition 1, we can easily prove (see Appendix B for the proof):

**Proposition 2.** *The concept influence for fairness (CIF) of a training sample $z_i^{tr}$ when counterfactually transformed to $\hat{z}_i^{tr}(c')$ based on the target concept $c'$ can be computed as:*

$$infl(D_{val}, \hat{\theta}_{i,c'}) \approx -\nabla_\theta \ell_{fair}(\hat{\theta})^\intercal H_{\hat{\theta}}^{-1} \left( \nabla\ell(z_i^{tr}; \hat{\theta}) - \nabla\ell(\hat{z}_i^{tr}(c'); \hat{\theta}) \right) \tag{10}$$

**Why Can CIF Improve Fairness?** We include the full theoretical analysis of why overriding training concepts using CIF framework can improve fairness in Appendix D. We briefly summarize here. When overriding label $Y$, we can change a training label of a disadvantaged group from a wrong label to a correct one, and effectively improve the performance of the model for this group. Therefore the label (re)assignment can reduce the accuracy disparities. Overriding sensitive attribute $A$ improves fairness by balancing the data distribution. Later in the experiments (Figure 14 in Appendix E.7), we show that the influence function often identifies the data from the majority group and recommends them to be changed to the minority group.

## 3 ALGORITHMIC DETAILS

We present our method of generating counterfactual samples and computing CIF.

### 3.1 GENERATING COUNTERFACTUAL SAMPLES

To compute the fairness influence based on Eqn. 10, we need to first generate the corresponding counterfactual sample $\hat{z}_i^{tr}(c') = (\hat{x}_i(c'), \hat{y}_i(c'), \hat{a}_i(c'), \hat{c}_i = c')$ when we override concept $C$ from $c$ to $c'$. Theoretically, generating the counterfactual examples requires the assumptions of the underlying causal graph but we use a set of practical algorithms to approximate.

**Overriding Label $Y$.** Since there is no variable in training data dependent on $Y$ (Figure 1c), we can simply change the sample's label to the target label $\hat{y}_i$ and keep other attributes unchanged, *i.e.* $\hat{z}_i^{tr}(\hat{y}_i) = (x_i, \hat{y}_i, a_i, \hat{c}_i = \hat{y}_i)$.

**Overriding Sensitive Attribute $A$.** When we override a sample's $A$, both its $X$ and $Y$ need to change (Figure 1a). This is the same as asking, *e.g.* in a loan application, "How a female applicant's profile (*i.e.* $x_i$) and the loan decision (*i.e.* $y_i$) would change, had she been a male (*i.e.* $a_i = \hat{a}_i$)?" Inspired by (Black et al., 2020), we train a W-GAN (Arjovsky et al., 2017) with *optimal transport mapping* (Villani et al., 2009) to generate *in-distributional*[5] counterfactual samples for $x_i$ as if $x_i$ belongs to a different $a_i$. To do so, we need to map the distribution of $X$ from $A = a$ to $A = a'$. We first partition the training samples' feature into two groups: $X|A = a$ and $X|A = a'$. Then we train a W-GAN with the generator $G_{a \to a'}$ as the approximated optimal transport mapping from $X|A = a$ to $X|A = a'$ and the discriminator $D_{a \to a'}$ ensures the mapped samples $G_{a \to a'}(X)$ and the real samples $X|A = a'$ are indistinguishable. The training objectives are the following:

$$
\begin{aligned}
\ell_{G_{a \to a'}} &= \frac{1}{n} \Big( \sum_{x \in X|A=a} D(G(x)) + \lambda \cdot \sum_{x \in X|A=a} c(x, G(x)) \Big) \\
\ell_{D_{a \to a'}} &= \frac{1}{n} \Big( \sum_{x' \in X|A=a'} D(x') - \sum_{x \in X|A=a} D(G(x)) \Big)
\end{aligned}
\tag{11}
$$

---

[5]We need the counterfactual samples to be in-distributional rather than out-of-distributional because we need the change between the counterfactual sample and the original sample to be large enough to impact the fairness measure. We tried counterfactual examples (Wachter et al., 2017) that impose minimum change to the original sample, and it does not work well in mitigation because the fairness influence value they induce is too small. Other approaches like data generation via causal graph only work on synthetic data.

where $n$ is the number of training samples, $\lambda$ is the weight balancing the conventional W-GAN generator loss (*i.e.* the first term in $\ell_{G_{a \to a'}}$) and the distance cost function $c(.)$ (*i.e.* $\ell_2$ norm in our case) that makes sure the mapped samples are not too far from the original distribution.

After we train the W-GAN on the training data, we can use the trained generator $G_{a \to a'}$ to map a sample $x_i$ to its counterfactual version $\hat{x}_i = G_{a_i \to \hat{a}_i}(x_i)$. In addition, once we have the counterfactual features, we can use the original model to predict the corresponding counterfactual label (*i.e.* following the dependency link $X \to Y$ in Figure 1a). The resulting counterfactual sample is $\hat{z}_i^{tr}(\hat{a}_i) = (\hat{x}_i, h_{\hat{\theta}}(\hat{x}_i), \hat{a}_i, \hat{c}_i = \hat{a}_i)$.

**Overriding Feature** $X$. In image data, assume there exists an image-label attribute $C = attr(X)$, *e.g.* young or old in facial images, and overriding $X$ means transforming the image (*i.e.* all pixel values in $X$) as if it belongs to a different $C$. In tabular data, $C$ is one of the features in $X$, and when $C$ is changed, all other features in $X$ need to change accordingly. In both cases, similar to overriding $A$, we train a W-GAN to learn the mapping from the group $X|C = c$ to $X|C = c'$; the resulting generator is $G_{c \to c'}$ and the generated counterfactual feature is $\hat{x}_i = G_{c_i \to \hat{c}_i}(x_i)$. Similarly, since the data dependency $X \to Y$ exists in our assumption in Figure 1b, we also use the original model's predicted label as the counterfactual label. The resulting counterfactual sample is $\hat{z}_i^{tr}(\hat{c}_i) = (\hat{x}_i, h_{\hat{\theta}}(\hat{x}_i), a_i, \hat{c}_i = \hat{x}_i)$.

**Removal.** Removing is simply setting the counterfactual sample to be null, *i.e.* $\hat{z}_i^{tr}(c') = \varnothing$.

## 3.2 COMPUTING INFLUENCE

Following (Koh & Liang, 2017), we use the Hessian vector product (HVP) to compute the product of the second and the third term in Eqn. 10 together. Let $v := \left( \nabla \ell(z_i^{tr}; \hat{\theta}) - \nabla \ell(\hat{z}_i^{tr}(c'); \hat{\theta}) \right)$, we can compute $H^{-1}v$ recursively (Agarwal et al., 2017):

$$\hat{H}_r^{-1}v = v + (I - \hat{H}_0)\hat{H}_{r-1}^{-1}v \tag{12}$$

where $\hat{H}_0$ is the Hessian matrix approximated on random batches. Let $t$ be the final recursive iteration, then the final CIF is $\text{infl}(D_{val}, \hat{\theta}_{i,c'}) \approx -\nabla_\theta \ell_{\text{fair}}(\hat{\theta})^\intercal \hat{H}_t^{-1}v$, where $\ell_{\text{fair}}(\hat{\theta})$ is the surrogate loss of fairness measure (*e.g.* Eqn. 4, 21 or 25).

Similar to (Koh & Liang, 2017), we assume the loss is twice-differentiable and strongly convex in $\theta$, so that $H_{\hat{\theta}}$ exists and is positive definite, *i.e.* $H_{\hat{\theta}}^{-1}$ exists. If the assumptions are not satisfied, the convergence would suffer. It is a well-documented problem in the literature (Koh & Liang, 2017; Basu et al., 2020a). However, our goal is not to propose a better influence function approximating algorithm; we aim to demonstrate the idea of leveraging influence function to help practitioners understand the unfairness. As better influence-approximating algorithms are invented, our framework is flexible enough to plug in and benefit from the improvement. Later in Section 4.1, we empirically show the accuracy of our influence estimation.

## 4 EXPERIMENTS

We present a series of experiments to validate the effectiveness of CIF in explaining and mitigating model unfairness, detecting biased/poisoned samples, and recommending resampling to balance representation.

We test CIF on 4 datasets: synthetic, COMPAS (Angwin et al., 2016), Adult (Kohavi et al., 1996), and CelebA (Liu et al., 2015). We report results on three group fairness metrics (DP, EOP, and EO, see Table 1 in Appendix C for the definition). We include dataset details in Appendix E.1 and experiment details in Appendix E.2.

## 4.1 MITIGATION PERFORMANCE

We test the CIF-based mitigation by first computing CIF values on all training samples, and then replacing samples with the highest CIF values by their corresponding generated counterfactual samples, and retraining the model. In each retraining on the removed training set, we repeat the training

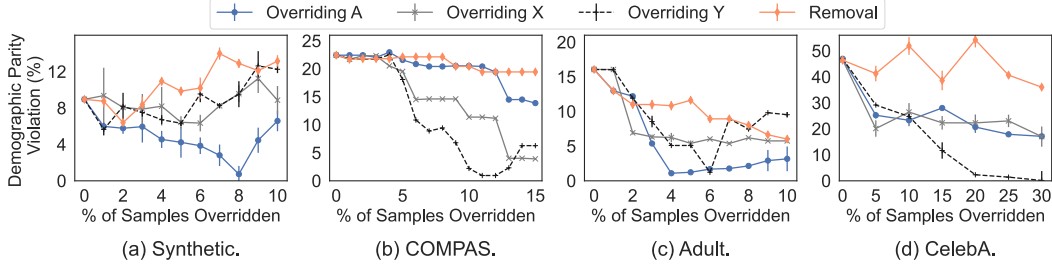

Figure 2: CIF-based mitigation performance with fairness measure Demographic Parity (DP).

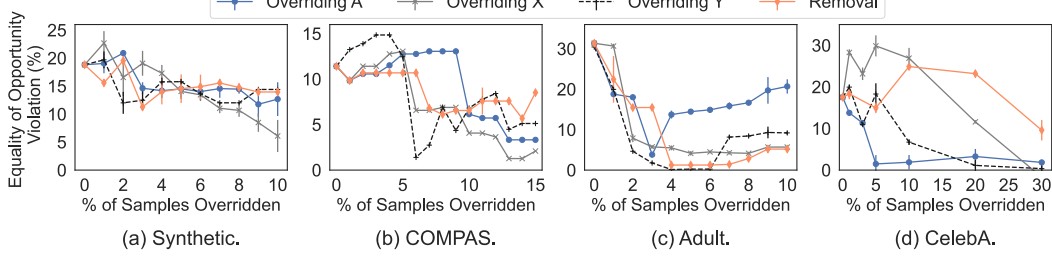

Figure 3: CIF-based mitigation performance with fairness measure Equality of Opportunity (EOP).

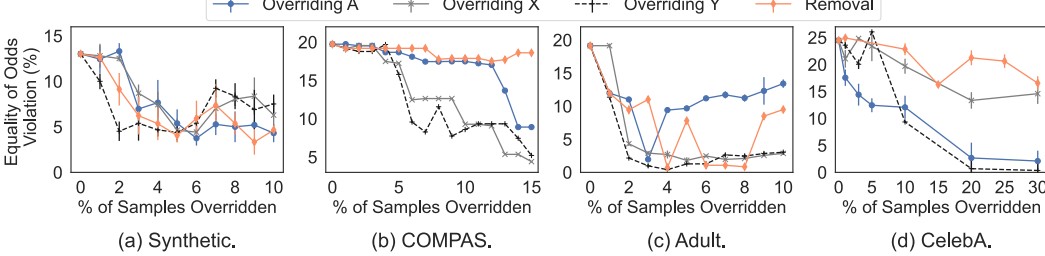

Figure 4: CIF-based mitigation performance with fairness measure Equality of Odds (EO).

process 10 times and report the standard deviation of the fairness measure in the error bars. Figure 2-4 show the fairness performance after the model training. We observe that all three fairness measures improve significantly after following CIF's mitigation recommendations. See Figure 8-10 in Appendix E.3 for the reported model accuracy.

We summarize observations: (1) Overriding $Y$ is highly effective on real-world data but not on synthetic. We conjecture that this is because we control the synthetic data to be cleanly labeled, which is not the case for other real-world data.[6] (2) Overriding $A$ proves to be helpful for most cases, especially for DP, which highly relates to the demographic variable $A$. (3) We set the size of synthetic data to be small (1,000) to show that simply removing training samples might not always be a good strategy, particularly on a small dataset in which the model would suffer significantly from losing training samples.

**Fairness-utility Tradeoff.** We report the fairness-utility tradeoffs of our mitigation on COMPAS, together with the in-processing mitigation (Agarwal et al., 2018) in Figure 5. Our mitigation is comparable to (Agarwal et al., 2018); sometimes we can achieve better fairness given a similar level of accuracy (*e.g.* when accuracy is $\sim 60\%$).

**Accuracy of CIF Estimate.** Figure 6 plots influence value vs. the actual difference in fairness loss (DP) on the COMPAS dataset. See Appendix E.5 for experiment details. The relationship between

---

[6]To test this hypothesis, we add label noise in the synthetic data to see if it would make Y-overriding more effective. Table 2 in Appendix E.4 shows the results. When labels are no long clean, our Y-overriding becomes more effective, showing that indeed label noise can be a significant contributor to the unfairness, as also indicated by prior work (Wang et al., 2021).

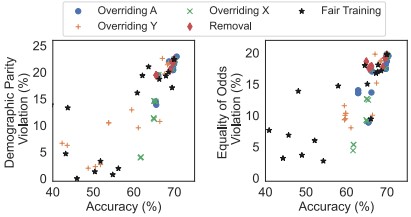

Figure 5: Fairness-accuracy tradeoff of CIF-based mitigation on COMPAS. CIF-based mitigation is comparable to in-processing mitigation method, and sometimes achieves better fairness given a similar level of accuracy.

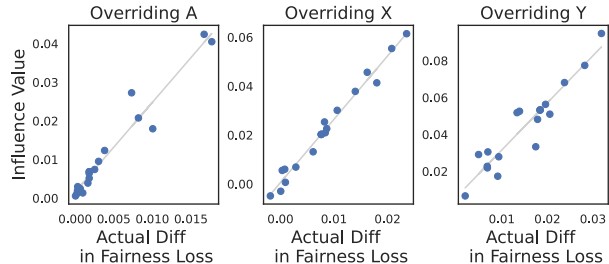

Figure 6: Estimated influence value vs. the actual difference in fairness loss on COMPAS with fairness metric Demographic Parity (DP).

our estimated influence and the actual change in fairness is largely linear, meaning our influence value can estimate the fairness change reasonably well.

**Distribution of CIF.** We show the distribution of influence values computed on COMPAS corresponding to three fairness metrics in Figure 11 (Appendix E.6). Overriding $Y$ has the highest influence value. This is because we change the value of $Y$ directly in this operation, which is more "unnaturally" compared to generating more "natural" counterfactual examples with W-GAN (overriding $X$ and $A$) or model-predicted value of $Y$ (overriding $X$). So practitioners should be particularly cautious about mislabelling, *e.g.* if any unprivileged group should be labeled favorable but ended up getting labeled unfavorable.

### 4.2 ADDITIONAL APPLICATIONS OF CIF

We provide three examples of additional applications that can be derived from our CIF framework: (1) fixing mislabelling, (2) defending against poisoning attacks, and (3) resampling imbalanced representations. We include experiment details and results in Appendix E.7.

**Fixing Mislabeling.** We flip training labels $Y$ in the Adult dataset to artificially increase the model's unfairness. Following (Wang et al., 2021), we add group-dependent label noise, *i.e.* the probability of flipping a sample's $Y$ is based on its $A$, to enlarge the fairness gap. We then compute $Y$-overriding CIF on each sample, and flag samples with the top CIF value. In Figure 12 (Appendix E.7), we report the precision of our CIF-based detection and mitigation performance if we flip the detected samples' labels and retrain the model. Our detection can flag the incorrect labels that are known to be the source of the unfairness with high precision (compared to randomly flagging the same percentage) and improves the model fairness if the detected labels are corrected.

**Defending against Poisoning Attacks.** We demonstrate the application of defending models against fairness poisoning attacks. To generate poisoned training samples that cause the model's unfairness, we choose poisoned training samples with the same probability based on the group- and label-dependent probability in the previous application. In addition to flipping the samples' labels, we also set the target feature (*i.e.* race in Adult) to be a fixed value (*i.e.* white) regardless of the original feature value. The attack that modifies a sample's feature to be a fixed value and changes its label is known as backdoor attack (Gu et al., 2019; Li et al., 2021b; Wu et al., 2022a), a special type of poisoning attack. After the poisoning, all fairness measures become worse. For detection, we compute $X$-overriding CIF on the poisoned feature, and flag samples with high CIF value. For mitigation, if we flag a sample to be poisoned, we remove it from the training set and retrain the model. Figure 13 (Appendix E.7) shows the precision of our detection and the mitigation performance after removal. We observe a high precision and reasonably good fairness improvement.

**Resampling Imbalanced Representations.** To create an extremely imbalanced representation in the training set, we upsample the positive samples in the privileged group (*i.e.* male) by $200\%$ in the Adult dataset, further increasing the percentage of positive samples that belong to the privileged group, and therefore the training samples are overwhelmingly represented by the privileged group.

The resulting fairness becomes worse. We then compute $A$-overriding CIF, and replace the high-influence samples with their counterfactual samples (*i.e.* adding counterfactual samples in the unprivileged group and reducing samples from the privileged group). In Figure 14 (Appendix E.7), we report the percentage of high-influence samples that belong to the privileged group (*i.e.* how much CIF recommends the data balancing) and the mitigation performance. The high-influence samples are almost all from the privileged group, which is expected, and if they were converted to the counterfactual samples as if they are from the unprivileged group, *i.e.* recollecting and resampling the training distribution, then fairness can improve.

## 5 RELATED WORK

**Influence Function.** The goal of influence function is to quantify the impact of training data on the model's output. (Koh & Liang, 2017) popularizes the idea of training data influence to the attention of our research community and has demonstrated its power in a variety of applications. Later works have aimed to improve the efficiency of computing influence functions. For example, Tracein (Pruthi et al., 2020) proposes a first-order solution that leverages the training gradients of the samples, and a neural tangent kernel approach for speeding up this task. Other works have explored the computation of group influence (Basu et al., 2020b), the robustness of influence function (Basu et al., 2020a), its application in explainable AI (Linardatos et al., 2020) and other tasks like graph networks (Chen et al., 2023).

**Influence Function for Fairness.** Our work is closely relevant to the recent discussions on quantifying training data's influence on a model's fairness properties. (Wang et al., 2022a) computes the training data influence to fairness when removing a certain set of training samples. (Li & Liu, 2022) discusses a soft version of the removal and computes also the optimal "removal weights" for each sample to improve fairness. And (Sattigeri et al., 2022) leverages the computed influence to perform a post-hoc model update to improve its fairness. Note that those works consider the fairness effect of removing or reweighing training samples. Our work targets a more flexible and powerful definition of influence that can give practitioners a wider scope of understanding by introducing the idea of concepts and generating counterfactual samples as well as result in a wider range of potential applications.

**Data Repairing for Fairness.** Our work is also related to the work on data repairing to improve fairness. (Krasanakis et al., 2018; Lahoti et al., 2020) discuss the possibilities of reweighing training data to improve fairness. (Zhang et al., 2022) proposes a "reprogramming" framework that modified the features of training data. (Liu & Wang, 2021) explores the possibility of resampling labels to improve the fairness of training. Other works study the robustness of model w.r.t fairness (Wang et al., 2022b; Chhabra et al., 2023; Li et al., 2022). Another line of research that repairs training data is through training data pre-proccessing (Calmon et al., 2017; Celis et al., 2020; Kamiran & Calders, 2012; du Pin Calmon et al., 2018), synthetic fair data (Sattigeri et al., 2019; Jang et al., 2021; Xu et al., 2018; van Breugel et al., 2021), and data augmentation (Sharma et al., 2020; Chuang & Mroueh, 2021).

## 6 CONCLUSIONS AND LIMITATIONS

We propose *Concept Influence for Fairness* (CIF), which generalizes the definition of influence function for fairness from focusing only on the effects of removing or reweighing the training samples to a broader range of dimensions related to the training data's properties. The main idea is to consider the effects of transforming the sample based on a certain *concept* of training data, which is a more flexible framework to help practitioners better understand unfairness with a wider scope and leads to more potential downstream applications.

We point out two limitations: (1) CIF needs to generate counterfactual samples w.r.t different concepts, which can be computationally expensive and (2) in CIF-based mitigation, it can be non-trivial to determine the optimal number of training samples to override that would maximally improve fairness.

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
