# A PROPOSITION 1: DERIVATION OF FAIRNESS FUNCTION ON GROUP FAIRNESS

Assume the risk of $\theta$ is $R(\theta) := \frac{1}{n} \sum_{i=1}^{n} \ell(z_i^{tr}; \theta)$, and the model trained on the entire training set is $\hat{\theta} := \operatorname{argmin}_\theta R(\theta)$. The resulting model weights if we assign weight $w_i \in [0, 1]$ to each sample $i \in \mathcal{K}$ and then upweight them by some small $\epsilon$ is the following:

$$\hat{\theta}_\mathcal{K} := \operatorname{argmin}_\theta \{R(\theta) + \epsilon \sum_{i \in \mathcal{K}} w_i \cdot \ell(z_i^{tr}; \theta)\}, \tag{13}$$

By the first order condition of $\hat{\theta}_\mathcal{K}$ we have

$$0 = \nabla R(\hat{\theta}_\mathcal{K}) + \epsilon \sum_{i \in \mathcal{K}} w_i \cdot \nabla \ell(z_i^{tr}; \hat{\theta}_\mathcal{K})$$

When $\epsilon \to 0$, with the Taylor expansion (and first-order approximation) we have:

$$0 \approx \left( \nabla R(\hat{\theta}) + \epsilon \sum_{i \in \mathcal{K}} w_i \cdot \nabla \ell(z_i^{tr}; \hat{\theta}) \right) + \left( \nabla^2 R(\hat{\theta}) + \epsilon \sum_{i \in \mathcal{K}} w_i \cdot \nabla^2 \ell(z_i^{tr}; \hat{\theta}) \right) \cdot (\hat{\theta}_\mathcal{K} - \hat{\theta})$$

By the first-order condition of $\hat{\theta}$ we have $\nabla R(\hat{\theta}) = 0$, and re-arranging terms we have

$$\frac{\hat{\theta}_\mathcal{K} - \hat{\theta}}{\epsilon} = - \left( H_{\hat{\theta}} + \epsilon \cdot \sum_{i \in \mathcal{K}} w_i \cdot \nabla^2 \ell(z_i^{tr}; \hat{\theta}) \right)^{-1} \cdot \left( \sum_{i \in \mathcal{K}} w_i \cdot \nabla \ell(z_i^{tr}; \hat{\theta}) \right)$$

Taking the limit of $\epsilon \to 0$ on both sides we have

$$\frac{\partial \hat{\theta}_\mathcal{K}}{\partial \epsilon} \bigg|_{\epsilon = 0} = -H_{\hat{\theta}}^{-1} \cdot \left( \sum_{i \in \mathcal{K}} w_i \cdot \nabla \ell(z_i^{tr}; \hat{\theta}) \right)$$

Finally, the fairness influence of assigning training sample $i$ in group $\mathcal{K}$ with weight $w_i$ is:

$$\operatorname{infl}(D_{val}, \mathcal{K}, \hat{\theta}) := \ell_{\text{fair}}(\hat{\theta}) - \ell_{\text{fair}}(\hat{\theta}_\mathcal{K}) \tag{14}$$

$$\approx \frac{\partial \ell_{\text{fair}}(\hat{\theta}_\mathcal{K})}{\partial \epsilon} \bigg|_{\epsilon = 0} \tag{15}$$

$$= \nabla_\theta \ell_{\text{fair}}(\hat{\theta})^\intercal \frac{\hat{\theta}_\mathcal{K}}{\partial \epsilon} \bigg|_{\epsilon = 0} \tag{16}$$

$$= -\nabla_\theta \ell_{\text{fair}}(\hat{\theta})^\intercal H_{\hat{\theta}}^{-1} \left( \sum_{i \in \mathcal{K}} w_i \nabla \ell(z_i^{tr}; \hat{\theta}) \right) \tag{17}$$

# B PROPOSITION 2: DERIVATION OF FAIRNESS FUNCTION FOR COUNTERFACTUAL SAMPLES

The proof follows largely from the one we presented above for Proposition 1 with the only difference being adapting the summation term to incorporate the addition of terms for counterfactual samples:

$$\epsilon \sum_{i \in \mathcal{K}} w_i \cdot \ell(z_i^{tr}; \theta)\} \to \epsilon \sum_{i \in \mathcal{K}} w_i \cdot \ell(z_i^{tr}; \theta)\} + \epsilon' \sum_{i \in \mathcal{K}} w_i' \cdot \ell(\hat{z}_i^{tr}; \theta)\}$$

and the counterfactual model now is defined as:

$$\hat{\theta}_\mathcal{K} := \operatorname{argmin}_\theta \{R(\theta) + \epsilon \sum_{i \in \mathcal{K}} w_i \cdot \ell(z_i^{tr}; \theta) + \epsilon \sum_{i \in \mathcal{K}} w_i' \cdot \ell(\hat{z}_i^{tr}; \theta)\}, \tag{18}$$

Similarly invoking the first-order condition we have

$$0 = \nabla R(\hat{\theta}_\mathcal{K}) + \epsilon \sum_{i \in \mathcal{K}} w_i \cdot \nabla \ell(z_i^{tr}; \hat{\theta}_\mathcal{K}) + \epsilon \sum_{i \in \mathcal{K}} w_i' \cdot \nabla \ell(\hat{z}_i^{tr}; \hat{\theta}_\mathcal{K})$$

which further offers us

$$\frac{\partial \hat{\theta}_{\mathcal{K}}}{\partial \epsilon}\Big|_{\epsilon=0} = -H_{\hat{\theta}}^{-1} \cdot \left( \sum_{i \in \mathcal{K}} w_i \cdot \nabla \ell(z_i^{tr}; \hat{\theta}) + \sum_{i \in \mathcal{K}} w_i' \cdot \nabla \ell(\hat{z}_i^{tr}; \hat{\theta}) \right)$$

and that

$$\text{infl}(D_{val}, \mathcal{K}, \hat{\theta}) := \ell_{\text{fair}}(\hat{\theta}) - \ell_{\text{fair}}(\hat{\theta}_{\mathcal{K}}) \approx -\nabla_\theta \ell_{\text{fair}}(\hat{\theta})^\intercal H_{\hat{\theta}}^{-1} \left( \sum_{i \in \mathcal{K}} w_i \nabla \ell(z_i^{tr}; \hat{\theta}) + \sum_{i \in \mathcal{K}} w_i' \nabla \ell(\hat{z}_i^{tr}; \hat{\theta}) \right) \tag{19}$$

By setting the the proper $w_i, w_i'$ (e.g., $w_i = \frac{1}{n}, w_i' = -\frac{1}{n}$) we recovered the claim made in Proposition 2.

## C APPROXIMATING FAIRNESS METRICS

Similarly to DP, we can approximate the violation of Equality of Opportunity (EOP) with:

$$\ell_{EOP}(\hat{\theta}) := \left| \mathbb{P}(h_\theta(X) = 1 | A = 0, Y = 1) - \mathbb{P}(h_\theta(X) = 1 | A = 1, Y = 1) \right| \tag{20}$$

$$\approx \left| \frac{\sum_{i \in D_{val}: a_i=0, y_i=1} g(z_i^{val}; \theta)}{\sum_{i \in D_{val}} \mathbb{I}[a_i = 0, y_i = 1]} - \frac{\sum_{i \in D_{val}: a_i=1, y_i=1} g(z_i^{val}; \theta)}{\sum_{i \in D_{val}} \mathbb{I}[a_i = 1, y_i = 1]} \right| \tag{21}$$

And for Equality of Odds (EO), we have

$$\ell_{EO}(\hat{\theta}) := \frac{1}{2} \Big( \left| \mathbb{P}(h_\theta(X) = 1 | A = 0, Y = 1) - \mathbb{P}(h_\theta(X) = 1 | A = 1, Y = 1) \right| + \tag{22}$$

$$\left| \mathbb{P}(h_\theta(X) = 1 | A = 0, Y = 0) - \mathbb{P}(h_\theta(X) = 1 | A = 1, Y = 0) \right| \Big) \tag{23}$$

$$\approx \frac{1}{2} \Big( \left| \frac{\sum_{i \in D_{val}: a_i=0, y_i=1} g(z_i^{val}; \theta)}{\sum_{i \in D_{val}} \mathbb{I}[a_i = 0, y_i = 1]} - \frac{\sum_{i \in D_{val}: a_i=1, y_i=1} g(z_i^{val}; \theta)}{\sum_{i \in D_{val}} \mathbb{I}[a_i = 1, y_i = 1]} \right| + \tag{24}$$

$$\left| \frac{\sum_{i \in D_{val}: a_i=0, y_i=0} g(z_i^{val}; \theta)}{\sum_{i \in D_{val}} \mathbb{I}[a_i = 0, y_i = 0]} - \frac{\sum_{i \in D_{val}: a_i=1, y_i=0} g(z_i^{val}; \theta)}{\sum_{i \in D_{val}} \mathbb{I}[a_i = 1, y_i = 0]} \right| \Big) \tag{25}$$

We summarize the definition and surrogate approximation of three group fairness measures as follows:

| Fairness Measure | Definition | Surrogate Approximation |
|---|---|---|
| Demographic Parity (DP) | $\left\| \mathbb{P}(h_\theta(X) = 1 \| A = 0) - \mathbb{P}(h_\theta(X) = 1 \| A = 1) \right\|$ | $\left\| \frac{\sum_{i \in D_{val}: a_i=0} g(z_i^{val}; \theta)}{\sum_{i \in D_{val}} \mathbb{I}[a_i=0]} - \frac{\sum_{i \in D_{val}: a_i=1} g(z_i^{val}; \theta)}{\sum_{i \in D_{val}} \mathbb{I}[a_i=1]} \right\|$ |
| Equality of Opportunity (EOP) | $\left\| \mathbb{P}(h_\theta(X) = 1 \| A = 0, Y = 1) - \mathbb{P}(h_\theta(X) = 1 \| A = 1, Y = 1) \right\|$ | $\left\| \frac{\sum_{i \in D_{val}: a_i=0, y_i=1} g(z_i^{val}; \theta)}{\sum_{i \in D_{val}} \mathbb{I}[a_i=0,y_i=1]} - \frac{\sum_{i \in D_{val}: a_i=1, y_i=1} g(z_i^{val}; \theta)}{\sum_{i \in D_{val}} \mathbb{I}[a_i=1,y_i=1]} \right\|$ |
| Equality of Odds (EO) | $\frac{1}{2} \left( \left\| \mathbb{P}(h_\theta(X) = 1 \| A=0, Y=1) - \mathbb{P}(h_\theta(X) = 1 \| A=1, Y=1) \right\| + \left\| \mathbb{P}(h_\theta(X) = 1 \| A=0, Y=0) - \mathbb{P}(h_\theta(X) = 1 \| A=1, Y=0) \right\| \right)$ | $\frac{1}{2} \left( \left\| \frac{\sum_{i \in D_{val}: a_i=0, y_i=1} g(z_i^{val}; \theta)}{\sum_{i \in D_{val}} \mathbb{I}[a_i=0,y_i=1]} - \frac{\sum_{i \in D_{val}: a_i=1, y_i=1} g(z_i^{val}; \theta)}{\sum_{i \in D_{val}} \mathbb{I}[a_i=1,y_i=1]} \right\| + \left\| \frac{\sum_{i \in D_{val}: a_i=0, y_i=0} g(z_i^{val}; \theta)}{\sum_{i \in D_{val}} \mathbb{I}[a_i=0,y_i=0]} - \frac{\sum_{i \in D_{val}: a_i=1, y_i=0} g(z_i^{val}; \theta)}{\sum_{i \in D_{val}} \mathbb{I}[a_i=1,y_i=0]} \right\| \right)$ |

Table 1: Fairness definition and surrogate approximation.

## D THEORETICAL ANALYSIS: WHY CAN CIF IMPROVE FAIRNESS?

**Overview.** We base the analysis on the data generation model adopted in (Feldman, 2020; Liu, 2021) to capture the impact of data patterns generated with different frequencies and the impact of label errors. This setup is a good fit for understanding how counterfactual data overriding can change the data frequency of different groups (majority group with higher frequency vs. minority group with lower frequency) and provides insights for CIF.

Overriding label $Y$ is relatively straightforward. If we are able to change a training label of a disadvantaged group from a wrong label to a correct one, we can effectively improve the performance

of the model for this group. Therefore the label (re)assignment can reduce the accuracy disparities. Our analysis also hints that the influence function is more likely to identify samples from the disadvantaged group with a lower presence in the data and mislabeled samples. This is because, for a minority group, a single label change would incur a relatively larger change in the influence value.

Overriding sensitive attribute $A$ improves fairness by balancing the data distribution. In the experiments (Figure 14), we show that the influence function often identifies the data from the majority group and recommends them to be changed to the minority group, as shown in Figure 7. In the analysis, we also show that this transformation incurs positive changes in the accuracy disparities between the two groups and therefore improves fairness.

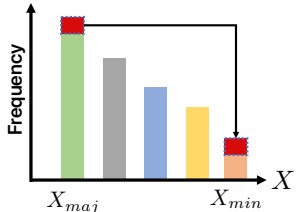

**Setup.** We base our analysis on the data generation model adopted in (Feldman, 2020; Liu, 2021) to capture the impact of data patterns generated with different frequencies and the impact of label errors. This setup is a good fit for understanding how counterfactual data overriding change the frequencies of data of different groups and therefore provides insights for CIF.

Figure 7: Illustration of the effect of overriding sensitive attribute $A$ as rebalancing data distribution.

In this setup, each feature $X$ takes value from a *discretized* set $\mathcal{X}$. For each $X \in \mathcal{X}$, sample a quantity $q_X$ independently and uniformly from a set $\lambda := \{\lambda_1, ..., \lambda_N\}$. The probability of observing an $X$ is given by $D(X) = q_X/(\sum_{X \in \mathcal{X}} q_X)$. Each $X$ is mapped to a true label $Y = f(X)$. But our observed training labels can be noisy, denoting as $\tilde{Y} \sim \mathbb{P}(\tilde{Y}|X, Y)$. $n$ pairs of $(X, \tilde{Y})$ are observed and collected for the dataset. Denote by $S_l$ the set of all samples that appear $l$ times in the dataset, and denote by $l[X]$ the number of appearances for $X$. Each $X$ is also associated with a sensitive group attribute $A$. Denote by $h_\theta$ as the classification model defined by $\theta \in \Theta$ (parametric space) and the generalization error over a given distribution $\mathcal{D}$ as

$$\text{err}_\mathcal{D}(h_\theta) := \mathbb{E}_\mathcal{D}[\mathbf{1}(h_\theta(X) \neq Y)] .$$

The following expected generalization error is defined in (Feldman, 2020):

$$\text{err}(\theta|D) := \mathbb{E}_{\mathcal{D} \sim \mathbb{P}[\cdot|D]} \left[ \text{err}_\mathcal{D}(h_\theta) \right] ,$$

where $\mathbb{P}[\cdot|D]$ is the distribution for the data distribution inferred from the dataset $D$. It is proved that:

**Theorem 1** ((Feldman, 2020)). $\text{err}(\theta|D) \geq \min_{\theta' \in \Theta} \text{err}(\theta'|D) + \sum_{l \in [n]} \tau_l \cdot \sum_{X \in S_l} \mathbb{P}[h_\theta(X) \neq Y]$.

In the above $\tau_l$ is a constant that depends on $l$. We call this the *importance* of an $l$-appearance sample. It is proven in (Feldman, 2020) that when $l$ is small, for instance $l = 1$, $\tau_l$ is at the order of $O(\frac{1}{n})$, and when $l$ is large $\tau_l$ is at the order of $O(\frac{l^2}{n^2})$ (Liu, 2021).

Consider an ideal setting where we train a parametric model $\theta$ that fully memorizes the training data that $R(\theta) = 0$, and therefore $\mathbb{P}[h_\theta(X) \neq Y] = \tilde{\mathbb{P}}[\tilde{Y} \neq Y|X]$, where $\tilde{\mathbb{P}}[\tilde{Y} \neq Y|X]$ is the empirical label distribution for sample pattern $X$. Theorem 1 can easily generalize to each group $D_a$:

**Proposition 3.** $\text{err}(\theta|D_a) \geq \min_{\theta' \in \Theta} \text{err}(\theta'|D_a) + \sum_{l \in [n]} \frac{\tau_l}{\sum_{X \in D_a} \tau_{l[X]}} \cdot \sum_{X \in D_a \cap S_l} \tilde{\mathbb{P}}[\tilde{Y} \neq Y|X]$.

Denote the following *excessive generalization error* for group $a$:

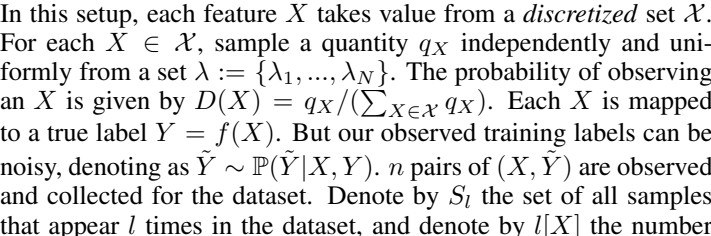

$$\text{err}_a^+(\theta|D) := \sum_{l \in [n]} \frac{\tau_l}{\sum_{X \in D_a} \tau_{l[X]}} \cdot \sum_{X \in D_a \cap S_l} \tilde{\mathbb{P}}[\tilde{Y} \neq Y|X].$$

Importantly, the above error term captures the vital quantities that are interesting to our problem: (1) the relevant frequency $\frac{\tau_l}{\sum_{X \in D_a} \tau_{l[X]}}$ captures the importance of the pattern with different frequencies and (2) $\tilde{\mathbb{P}}[\tilde{Y} \neq Y|X]$ the label noise rate of sample pattern $X$.

To set up the discussion, suppose we have two groups $a, a'$. $a$ is the advantaged group with a smaller $\text{err}_a^+(\pi, \theta|D)$; there is an $X_a \in D_a$ with a larger $l_a$. On the other hand, there is an $X_{a'} \in D_{a'}$, an

$l_{a'}$-appearance sample. We further assume that $l_a > l_{a'}$ ($X_{a'}$ has a lower representation). The rest of the discussion will focus on the following generalization error disparity as the fairness metric:

$$F(\theta) := |\text{err}_a^+(\theta|D) - \text{err}_{a'}^+(\theta|D)| \ .$$

The excessive generalization error for each group can be viewed as the expected influence of a model $\theta$ on the test data for that particular group. So the rest of the analysis focuses on the impact of flipping a sample's label to the group's excessive generalization error and then $F(\theta)$.

**Overriding Labels** ($Y$). On the high level, overriding a wrong label from the disadvantaged group $a'$ to the correct one will effectively reduce $\tilde{\mathbb{P}}[\tilde{Y} \neq Y | X]$ for some $X \in D_{a'}$, and therefore reduces the gap from it to the advantaged groups. The literature on influence functions (Koh & Liang, 2017) has demonstrated its power to detect mislabelled samples. But why would the influence function identify samples from the disadvantaged group and samples with wrong labels?

Consider a specific sample $X_{a'} \in D_{a'}$, and suppose its label is wrong. Overriding the wrong label to the correct label for this rare sample leads to a reduction in noise rate $\tilde{\mathbb{P}}[\tilde{Y} \neq Y | X]$ for $X_{a'}$. Therefore we know that overriding this "rare sample" reduces $\text{err}_{a'}^+(\theta|D)$, and the disparity $F(\theta)$. On the other hand, overriding the label for $X_a$ from the privileged group reduces $\text{err}_a^+(\theta|D)$ but this would further increase the gap $F(\theta)$. Therefore, flipping (*i.e.* overriding) the wrong labels from the disadvantaged group leads to a larger drop in disparity.

**Overriding Sensitive Attributes** ($A$). Suppose $X_a \in D_a$ (from the privileged group) is identified to be overridden. After the counterfactual overriding, $X_a$ is overridden to $X_{a'}$ (from the disadvantaged group), we show the gap in the excessive generalization errors between $a$ and $a'$ is reduced as follows:

*(1) Increase in generalization error for the privileged group*: For group $a$'s generalization error, since we are removing one sample from it, the *importance* of $X_a$ drops from $\tau_{l_a}$ to $\tau_{l_a-1}$ as $\tau_l$ monotonically increases w.r.t $l$ (recall $\tau_l$ implies the importance of a $l$-frequency sample, the higher $l$ is the more important it generally is). When $X_a$ is a cleaner example that $\tilde{\mathbb{P}}[\tilde{Y} \neq Y | X_a]$ is sufficiently small, especially smaller than the average noise rate $\tilde{\mathbb{P}}[\tilde{Y} \neq Y | X \in D_a]$ of the group $a$, removing one sample of it results in an increase in the average generalization error (Proposition 4).

*(2) Decrease in generalization error for the privileged group*: For group $a'$, because of the addition, the weight of $X_{a'}$ increases by $\tau_{l_{a'}+1} - \tau_{l_{a'}}$. Therefore, adding a cleaner sample to group $a'$ not only reduces $X_{a'}$'s empirical label noise rate $\tilde{\mathbb{P}}[\tilde{Y} \neq Y | X_a]$, but also increases the relative weight of $\tau_{l_{a'}}$. Again using Proposition 4, we know that increasing the weight of a smaller quantity will then reduce the average of the group.

To summarize the above, overriding $A$ effectively (1) increases $\text{err}_a^+(\theta|D)$ (*i.e.* increasing the excessive generalization error for the privileged group) and (2) decreases $\text{err}_{a'}^+(\theta|D)$ (*i.e.* decreasing the excessive generalization error for the disadvantaged group). Therefore the counterfactual overriding $A$ reduces the gaps in the excessive generalization errors between the two groups.

### D.1 PROOF OF PROPOSITION 3

Recall we assume a simplified case where we train a parametric model $\theta$ that fully memorizes the training data that $R(\theta) = 0$, and therefore $\mathbb{P}[h_\theta(X) \neq Y] = \tilde{\mathbb{P}}[\tilde{Y} \neq Y | X]$. Following the proof from (Feldman, 2020), it is easy to show that

$$\mathbb{E}_{\mathcal{D} \sim \mathbb{P}[\cdot|D]}\left[\mathbb{P}_{\mathcal{D}}(h_\theta(X) \neq Y, X \in D_a)\right] \geq \min_{\theta' \in \Theta} \text{err}(\theta', X \in D_a) + \sum_{l \in [n]} \tau_l \cdot \sum_{X \in D_a \cap S_l} \tilde{\mathbb{P}}[\tilde{Y} \neq Y | X]$$

This is done simply by restricting generalization error to focus on data coming from a particular subset $D_a$. Note that

$$\mathbb{E}_{\mathcal{D} \sim \mathbb{P}[\cdot|D]}\left[\mathbb{P}_{\mathcal{D}}(h_\theta(X) \neq Y, X \in D_a)\right] = \mathbb{E}_{\mathcal{D} \sim \mathbb{P}[\cdot|D]}\left[\mathbb{P}_{\mathcal{D}}(h_\theta(X) \neq Y | X \in D_a) \cdot \mathbb{P}_{\mathcal{D}}(X \in D_a)\right]$$

Assuming the independence of the samples drawn, we have

$$\mathbb{E}_{\mathcal{D} \sim \mathbb{P}[\cdot|D]}\left[\mathbb{P}_{\mathcal{D}}(h_\theta(X) \neq Y, X \in D_a)\right] = \mathbb{E}_{\mathcal{D} \sim \mathbb{P}[\cdot|D]}\left[\mathbb{P}_{\mathcal{D}}(h_\theta(X) \neq Y | X \in D_a)\right]$$
$$\cdot \mathbb{E}_{\mathcal{D} \sim \mathbb{P}[\cdot|D]}\left[\mathbb{P}_{\mathcal{D}}(X \in D_a)\right]$$

From the above, we derive that

$$\mathbb{E}_{\mathcal{D}\sim\mathbb{P}[\cdot|D]}\left[\mathbb{P}_{\mathcal{D}}(h_\theta(X)\neq Y|X\in D_a)\right] = \frac{\mathbb{E}_{\mathcal{D}\sim\mathbb{P}[\cdot|D]}\left[\mathbb{P}_{\mathcal{D}}(h_\theta(X)\neq Y, X\in D_a)\right]}{\mathbb{E}_{\mathcal{D}\sim\mathbb{P}[\cdot|D]}\left[\mathbb{P}_{\mathcal{D}}(X\in D_a)\right]}. \quad (26)$$

According to the definition of $\tau$ in (Feldman, 2020) we have

$$\mathbb{E}_{\mathcal{D}\sim\mathbb{P}[\cdot|D]}\left[\mathbb{P}_{\mathcal{D}}(X\in D_a)\right] = \mathbb{E}_{\mathcal{D}\sim\mathbb{P}[\cdot|D]}\left[\sum_{X\in D_a}\mathcal{D}(X)\right]$$

$$= \sum_{X\in D_a}\mathbb{E}_{\mathcal{D}\sim\mathbb{P}[\cdot|D]}\left[\mathcal{D}(X)\right]$$

$$= \sum_{X\in D_a}\tau_{l[X]} \quad \text{(Definition of } \tau)$$

Plugging the above back into Eqn 26 gives

$$\mathsf{err}(\theta|D_a) \geq \min_{\theta'\in\Theta}\mathsf{err}(\theta'|D_a) + \sum_{l\in[n]}\frac{\tau_l}{\sum_{X\in D_a}\tau_{l[X]}}\cdot\sum_{X\in D_a\cap S_l}\tilde{\mathbb{P}}[\tilde{Y}\neq Y|X].$$

### D.2   BASIC THEOREM FOR PROPOSITION 4

We next prove the following:

**Proposition 4.** *For a set of non-negative numbers $\{b_1,...,b_N\}$ with their associated non-negative weights $\{w_1,...,w_N\}$ such that $\sum_{i=1}^N w_i = 1$. Denote the average as $\bar{b}:=\sum_{i=1}^N w_i b_i$. Then*

*(1) For $b_i < \bar{b}$, change its weight from $w_i$ to $w_i' < w_i$, and every other weight stays unchanged s.t. $w_j' = w_j$. Given the following renormalization $w_j' = \frac{w_j'}{\sum_i w_i'}$, we have $\bar{b}':=\sum_i w_i' b_i > \bar{b}$.*

*(2) For any particular $b_i < \bar{b}$, change its $b_i$ to $b_i' < b_i$ and keep other $b_j, j\neq i$ unchanged that $b_j' = b_j$. Furthermore, change its weight from $w_i$ to $w_i' > w_i$, and every other weight stays unchanged s.t. $w_j' = w_j$. Given the following renormalization $w_j' = \frac{w_j'}{\sum_i w_i'}$, we have $\bar{b}':=\sum_i w_i' b_i < \bar{b}$.*

*Proof.* To prove (1), we have

$$\bar{b}' - \bar{b} = \sum_j (w_j' - w_j)\cdot b_j$$

$$= \sum_{j\neq i}(w_j' - w_j)\cdot b_j + \left((1-\sum_{j\neq i}w_j') - (1-\sum_{j\neq i}w_j)\right)b_i$$

$$= \sum_{j\neq i}(w_j' - w_j)\cdot(b_j - b_i)$$

Furthermore, let $\Delta = w_i - w_i'$, for $j\neq i$ we have:

$$w_j' - w_j = \frac{w_j}{1-\Delta} - w_j = w_j\cdot\frac{\Delta}{1-\Delta}$$

Therefore we have

$$\sum_{j\neq i}(w_j' - w_j)\cdot(b_j - b_i) = \frac{\Delta}{1-\Delta}\cdot\sum_{j\neq i}w_j(b_j - b_i)$$

$$= \frac{\Delta}{1-\Delta}\cdot\left((\bar{b} - w_i\cdot b_i) - (b_i - w_i\cdot b_i)\right)$$

$$= \frac{\Delta}{1-\Delta}\cdot(\bar{b} - b_i) > 0$$

To prove (2), we basically follow the same proof. The only difference is that now let $\Delta = w_i' - w_i$, then for $j \neq i$:

$$w_j' - w_j = \frac{w_j}{1 + \Delta} - w_j = -w_j \cdot \frac{\Delta}{1 + \Delta}$$

Then we have

$$
\begin{aligned}
\bar{b}' - \bar{b} &= \sum_j (w_j' - w_j) \cdot b_j + w_i' \cdot (b_i' - b_i) \\
&= \sum_{j \neq i} (w_j' - w_j) \cdot (b_j - b_i) + w_i' \cdot (b_i' - b_i) \\
&= -\frac{\Delta}{1 + \Delta} \cdot \sum_{j \neq i} w_j (b_j - b_i) + w_i' \cdot (b_i' - b_i) \\
&= -\frac{\Delta}{1 + \Delta} \cdot \left( (\bar{b} - w_i \cdot b_i) - (b_i - w_i \cdot b_i) \right) + w_i' \cdot (b_i' - b_i) \\
&= -\frac{\Delta}{1 + \Delta} \cdot (\bar{b} - b_i) + w_i' \cdot (b_i' - b_i) < 0
\end{aligned}
$$

$\square$

# E   ADDITIONAL EXPERIMENTAL RESULTS AND DETAILS

## E.1   DATASET DETAILS

We include the details of datasets in the following:

- **Synthetic**: We generate synthetic data with the assumed causal graphs in Figure 1, and therefore we have the ground-truth counterfactual samples. See Appendix E.1 for the dataset generation process. Model: logistic regression.
- **COMPAS**: Recidivism prediction data (we use the preprocessed tabular data from IBM's AIF360 toolkit (Bellamy et al., 2019)). Feature $X$: tabular data. Label $Y$: recidivism within two years (binary). Sensitive attribute $A$ (removed from feature $X$): race (white or non-white). Model: logistic regression. When overriding $X$, we choose to flip the binary feature (age $> 45$ or not) in $X$.
- **Adult**: Income prediction data (we use the preprocessed tabular data from IBM's AIF360 toolkit (Bellamy et al., 2019)). Feature $X$: tabular data. Label $Y$: if income $> 50K$ or not. Sensitive attribute $A$ (removed from feature $X$): sex (male or female). Model: logistic regression. When overriding $X$, we choose to flip the binary feature race (white or non-white) in $X$.
- **CelebA**: Facial image dataset. Feature $X$: facial images. Label $Y$: attractive or not (binary). Sensitive attribute $A$: gender (male and female). Model: ResNet18 (He et al., 2016). When overriding $X$, we choose to flip the binary image-level label "Young."

The synthetic data is generated using a DAG with specified equations as follows:

$$
\begin{aligned}
X_1 &\sim \text{Normal}(0, 1) \\
A &\sim \text{Bernoulli}(0.3) \\
X_2 &\sim \text{Normal}(A, 3) \\
Z_1 &\sim \text{Normal}(0, 1) \\
X_3 &\sim \text{Normal}(2 \cdot Z_1 - 1, 0.1) \\
X_4 &\sim \text{Bernoulli}(0.1) \\
Y &= sign(5 \cdot X_1 \cdot A + 0.2 \cdot X_2^3 + 0.5 \cdot A + 0.3 \cdot X_4 - X_3)
\end{aligned}
$$

We use $X_1, X_2, X_3, X_4, A$ as features, $A$ as sensitive attributes, and $Y$ as labels.

We split all tabular datasets randomly into $70\%$ training, $15\%$ validation, and $15\%$ test set. We use the original data splitting in CelebA.

## E.2 EXPERIMENT DETAILS

We train the logistic regression on synthetic, Adult, and COMPAS using SGD with a learning rate 0.01. For CelebA, we train ResNet18 using Adam with a learning rate 0.001.

**Generating Image Counterfactual Samples.** When generating image counterfactual samples, we find directly using the generated images from W-GAN does not lead to a satisfactory mitigation performance because the distance between the counterfactual sample and the original sample is too small to impose a change that is large enough to improve fairness (tabular data has no such problem). Therefore we use a heuristic in CIF-based mitigation for image data. Using overriding $X$ as the example, when we map a sample's feature from $X|C = c$ to $X|C = c'$, we get the counterfactual feature $\hat{x}_i = G_{c_i \to \hat{c}_i}(x_i)$. We then search from the real examples $X|C = c'$ to find the nearest neighbor (in the original model's feature space) of $\hat{x}_i$, *i.e.*

$$\hat{x}_i^{'} = \operatorname*{arg\,min}_{x \sim X|C=c'} ||g_{\hat{\theta}}(\hat{x}_i) - g_{\hat{\theta}}(x)||^2 \tag{27}$$

where $g_{\hat{\theta}}$ is the feature extractor of the original model. That is to say, we search from the pool of real samples belonging to the target group closest to the generated fake sample. Since now the counterfactual feature is another real sample in the training data, it is directly removing a sample and replace with another real sample, which induces a larger change than replacing with a fake sample that needs to be reasonably close to the original sample in the W-GAN's training constraint. The resulting counterfactual sample is $\hat{z}_i^{tr}(\hat{c}_i) = (\hat{x}_i^{'}, h_{\hat{\theta}}(\hat{x}_i^{'}), a_i, \hat{c}_i)$. In experiments, we cap the nearest neighbor search space to be $10\%$ of the target group size to reduce the computational cost.

## E.3 ADDITIONAL MITIGATION RESULTS

Figure 8-10 show the model accuracy after applying CIF-based mitigation.

## E.4 IMPACT OF LABEL NOISE

We add label noise to the synthetic data, and Table 2 shows the effectiveness of Y-overriding when we override 10% top samples flagged by our method.

| Noisy Rate | 0% | 5% | 10% | 15% | 20% | 25% |
|---|---|---|---|---|---|---|
| Demographic Disparity | 12.4% | 2.1% | 6.5% | 1.7% | 0.05% | 2.2% |

Table 2: Effectiveness of Y-overriding on synthetic data when label noise exists.

## E.5 ACCURACY OF ESTIMATED INFLUENCE VALUE

Figure 6 plots influence value vs. the actual difference in fairness loss (DP) on COMPAS dataset. For the first data point, we remove 100 training samples with the largest influence value from the training set, retrain the model, and compute the actual change of fairness loss (eq.4) from the original model. For the next data point, we pick samples with the next 100 largest influence values and so on. We can see the relationship between our estimated influence and the actual change in fairness is largely linear, meaning our influence value can estimate the fairness change reasonably well.

## E.6 DISTRIBUTION OF INFLUENCE VALUES

Figure 11 shows the distribution of influence values computed on COMPAS corresponding to three fairness metrics.

## E.7 DETAILS OF EXPERIMENTS ON ADDITIONAL APPLICATIONS

**Fixing Mislabelling.** The group-dependent label noise rate we add to the Adult training dataset is shown in Table 3. We follow a similar experimental setting in (Wang et al., 2021). After the label

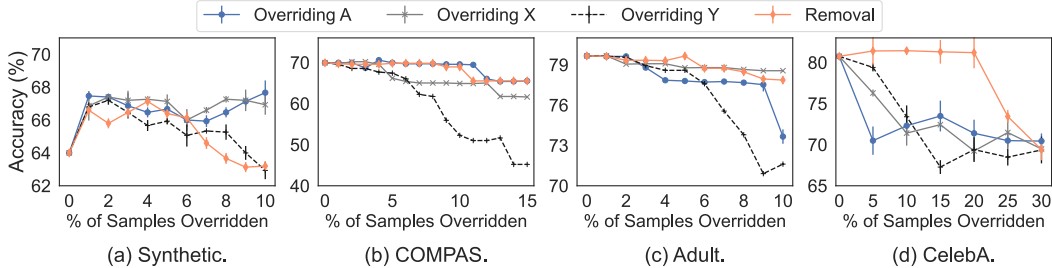

Figure 8: Model accuracy with CIF-based mitigation using fairness measure Demographic Parity (DP).

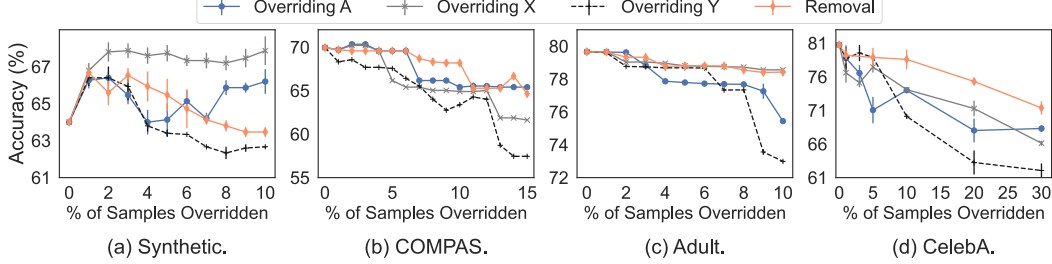

Figure 9: Model accuracy with CIF-based mitigation using fairness measure Equality of Opportunity (EOP).

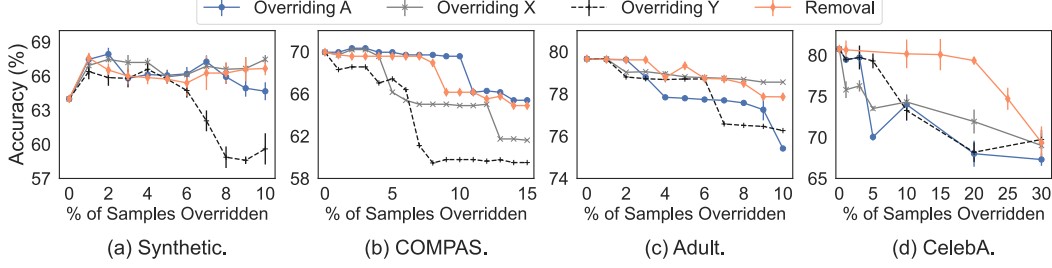

Figure 10: Model accuracy with CIF-based mitigation using fairness measure Equality of Odds (EO).

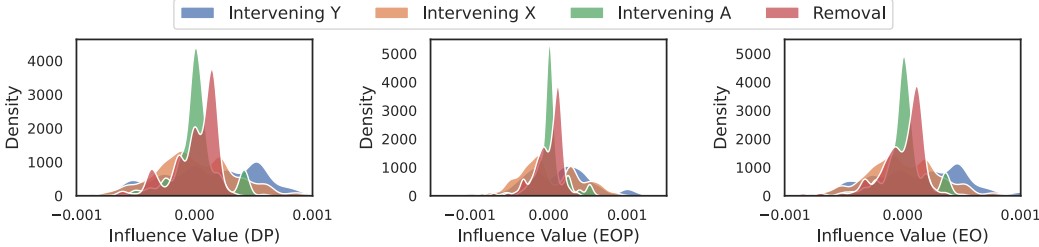

Figure 11: Distribution of influence values computed on COMPAS across three fairness metrics.

|       | A = 0 | A = 1 |
|-------|-------|-------|
| Y = 0 | 0.45  | 0.35  |
| Y = 1 | 0.15  | 0.55  |

Table 3: Group-dependent label noise rate added in the training samples in Adult data.

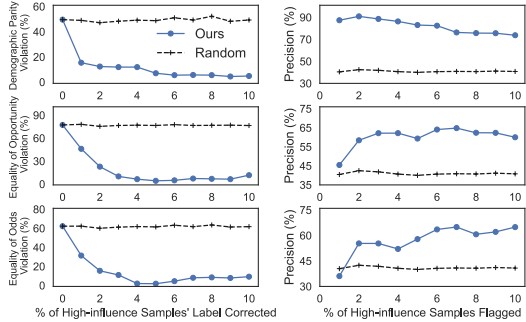

Figure 12: Precision and mitigation performance of using $Y$-overriding CIF to detect and correct training mislabeling that causes bias on Adult.

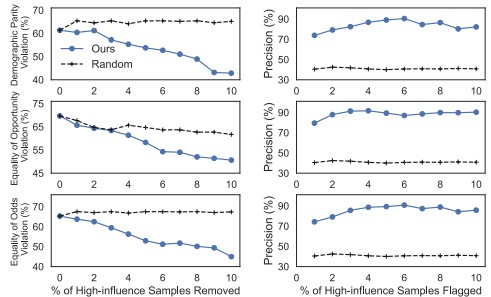

Figure 13: Precision and mitigation performance of using $X$-overriding CIF to detect and correct poisoned training samples that cause unfairness on Adult.

Figure 14: Performance of using $A$-overriding CIF to detect and correct imbalanced training representation that causes unfairness on Adult.

overriding, the bias increases significantly: DP increases from $16.1\%$ to $49.6\%$, EOP increases from $31.4\%$ to $77.3\%$, and EO increases from $19.1\%$ to $63.3\%$.

We flag samples by choosing samples with top influence when $Y$ is overridden and report the precision ($\frac{\#\text{flipped labels correctly detected}}{\#\text{flagged labels}}$) of our detection.

**Defending against Poisoning Attacks.** After the training samples are poisoned, the model unfairness increases as follows: DP increases from $16.1\%$ to $61.4\%$, EOP increases from $31.4\%$ to $69.4\%$, and EO increases from $19.1\%$ to $65.2\%$.

**Resampling Imbalanced Representations.** After artificially unbalancing the training samples, the fairness gap increases as follows: DP increases from $16.1\%$ to $42.6\%$, EOP increases from $31.4\%$ to $63.7\%$, and EO increases from $19.1\%$ to $47.2\%$.

### E.8 GENERATED COUNTERFACTUAL SAMPLES

Figure 15 and 16 show some random examples of generated images in CelebA when overriding $A$.

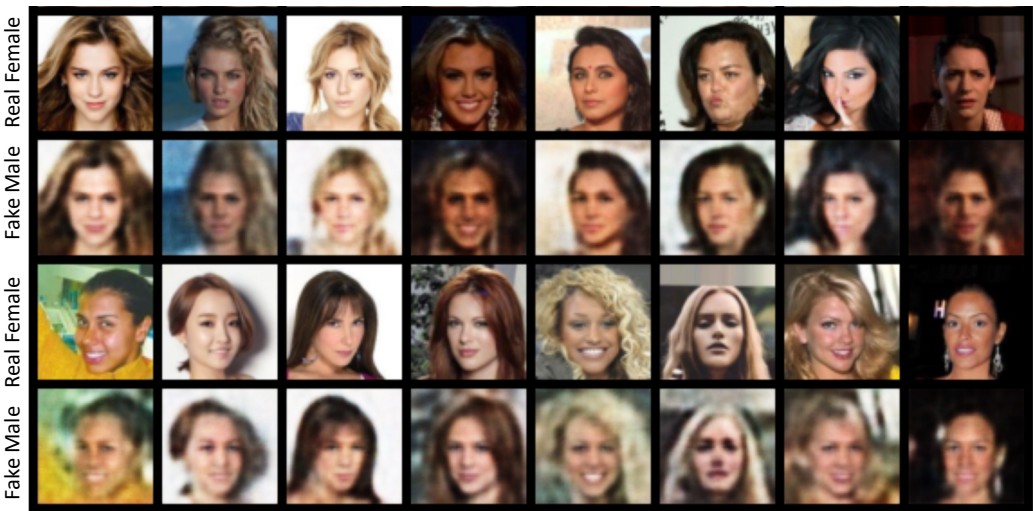

Figure 15: W-GAN generated images that map from male to female in CelebA.

Figure 16: W-GAN generated images that map from female to male in CelebA.