# OpenReview forum: "Understanding Unfairness via Training Concept Influence"
_ICLR.cc/2024/Conference — Submitted to ICLR 2024_

### Official Review · Reviewer_AAoD · 2023-10-31

**Soundness:** 2 fair
**Presentation:** 4 excellent
**Contribution:** 3 good
**Rating:** 6
**Confidence:** 4

**Summary:**

This paper aims to identify the causes of a model’s unfairness by leveraging the method of influence function. Instead of only detecting the training samples that induce unfairness directly and passively, the article takes a step further and tries actively to replace the “high-influence/biased” samples with new counterfactual samples generated by designing an algorithm. This solution is interesting and intuitive. In addition, the framework proposed in this paper is general and has the potential to be applied to various new scenarios.

**Strengths:**

1. this paper is well-written;
2. the topic is both intriguing and of practical significance.
3. the main idea is intuitive and easy to understand;
4. the algorithm design is reasonable.

**Weaknesses:**

1. There are no theoretical guarantees about the methods used to generate counterfactual samples.

**Questions:**

I do not have any major concerns about this work in technical details or presentation.

---

> ### Author Response · Authors · 2023-11-15
> **Rebuttal**
>
> We thank the reviewer for considering our work to be "well-written," "intriguing," "practical," "intuitive," "easy to understand", and  "reasonable design."
>
> We address the following concern.
>
> ### **1. Theoretical Understanding**
> We have dedicated an entire section on the theoretical insights in Appendix D. We did not include it in the main paper due to the page limit.
>
> We briefly summarize here why intervening training data attributes using CIF framework can improve fairness. For simplicity, we focus on accuracy disparity as the fairness measure.
>
> We base the analysis on the data generation model adopted in [1] to capture the impact of data patterns generated with different frequencies and the impact of label errors. This setup is a good fit for understanding how counterfactual data interventions can change the data frequency of different groups (majority group with higher frequency vs. minority group with lower frequency) and provides insights for CIF.
>
> Intervening labels $Y$ is relatively straightforward. If we are able to intervene on a training label of a disadvantaged group from a wrong label to a correct one, we can effectively improve the performance of the model for this group. Therefore the label intervention can reduce the accuracy disparities. Our analysis also hints that
> the influence function is more likely to identify samples from the disadvantaged group with a lower presence in the data and mislabeled samples. This is because, for a minority group, a single label change would incur a relatively larger change in the influence value.
>
> Intervening sensitive attributes $A$ improves fairness by ``balancing" the data distribution. In the experiments, we show that the influence function often identifies the data from the majority group and recommends them to be intervened to the minority group. In the analysis, we also show that this intervention incurs positive changes in the accuracy disparities between the two groups and therefore improves fairness.
>
> Please let us know if there is still anything unclear.
>
>
> [1] Feldman, Vitaly. "Does learning require memorization? a short tale about a long tail." Proceedings of the 52nd Annual ACM SIGACT Symposium on Theory of Computing. 2020.

---

> > ### Comment · Reviewer_AAoD · 2023-11-19
> > **Thanks for the response**
> >
> > Thanks for the response. I have no further questions and tend to remain the score.

---

### Official Review · Reviewer_rxAc · 2023-11-01

**Soundness:** 3 good
**Presentation:** 2 fair
**Contribution:** 2 fair
**Rating:** 5
**Confidence:** 3

**Summary:**

This paper introduces tools from the influence function literature to quantify algorithmic fairness.

**Strengths:**

See below for contextual discussion of strengths.

**Weaknesses:**

This paper uses influence functions to quantify fairness. The topic is one of the paper's strengths; although influence functions and counterfactual fairness models are well-established, the connection between the two is (to my knowledge) less developed. At least for me, for a paper combining two well-worn topics, it is important that the paper does so in an exceptionally clear or general way. The paper has potential for doing so but may be somewhat lacking in its current framing (at least according to my current understanding of the paper and contextual literature). For example, much of the discussion seems to assume a relatively limited class of ML problems (e.g., binary classification). Also, the relationship between ''Concepts'' and ''Fairness'' could (in my view) use a clearer elaboration. ''Transforming the sample based on certain concepts of training data'' seems far more general (this could concern any number of modifications). This framing somewhat complicates (at least this readers') analysis of the paper. I would consider revisions that streamline some of this. The introduction of the override operator also wasn't particularly obvious to me (e.g., ''override() operator counterfactually sets the value of the concept variable to a different c0.'' sounds like the do() operation, although I see an attempt to distinguish).

Overall, I see promise in the paper, but (at least in my opinion), certain limitations exist in terms of clarity. Also, the methods don't seem to have a comparison benchmark (even a naive one) which would help readers understand the contribution over any existing approaches in this context.

A few more focused comments.

(1) The paper uses the term ''fairness'' throughout. It would be helpful to provide an explicit definition early on. At times, it seems like the term is more qualitative (e.g., ''Influence Functions for Fairness''; ''They can all contribute to unfairness''). At other times, the term seems to be in some sense quantitative (i.e., ''fairness can improve''; ''fairness becomes worse''). Although the meaning may be clear to readers embedded in the fair ML literature, I think it would help to fix ideas early on for readers. Having said that, what constitutes ''fair'' (in at least some of its meanings) is, of course, subject to social values.

(2) It could help to discuss some of the challenges of computing the Hessian vector product (e.g., in high dimensions) and to emphasize why bypassing the Hessian calculation via the HVP can be useful. Would the influence function approach break down, e.g., when the parameter number ranges in the billions? More generally, based on the paper's existing framing, I had a hard time understanding some of its limitations, which are not particularly emphasized in the paper (the ''Conclusions and Limitations'' section is 10 lines long.)

**Questions:**

(1) Could the authors clarify what is meant by, ''we show that the influence function often identifies the data from the majority group and recommends them to be changed to the minority group''? Presumably, this refers to re-weighting of the training data, but it reads almost like the direct changing/manipulation of observed data values.

(2) Do all of the performance metrics have definitions in the paper? (I don't see one, for example, for the ''Equality of Odds'' metric). If the metrics could be compiled into a table with equations, this reader may have a better understanding of them and the results.

**Details Of Ethics Concerns:**

No major ethnics concerns.

---

> ### Author Response · Authors · 2023-11-15
> **Rebuttal Part 1**
>
> We thank the reviewer for considering our problem to be important. We address the concerns one by one.
>
> ### **1. Applicability**
> We thank the reviewer for the concern. Although we test binary classification models in our experiments, our method is flexible enough to be extended to multi-class fairness - the only difference is the definition of fairness, which is easy to plug in.
>
> We admit that extending to the regression or generative models would require significant change, but almost all works in influence function (we mean general influence function, not even the influence function for fairness) only target the classification models, and therefore we think our scope is reasonable.
>
> ### **2. Relationship between "Concepts" and "Fairness"**
> We thank the reviewer for asking for clarification.
>
> In our case, the concept can be sensitive attribute $A$, feature $X$, or label $Y$. Consider group fairness like Demographic Parity (DP): $\big| \mathbb{P}(h_\theta(X) = 1 | A = 0)-\mathbb{P}(h_\theta(X) = 1 | A = 1) \big|$, we can see it depends on feature $X$ and sensitive attribute $A$, and therefore the concept $X$ and $A$ are closely related to the fairness. Or consider Equality of Opportunity (EOP): $\big| \mathbb{P}(h_\theta(X) = 1 | A = 0, Y=1)-\mathbb{P}(h_\theta(X) = 1 | A = 1, Y=1) \big|$, it depends on all $X$, $Y$, $A$.
>
> However, the relationship cannot be derived precisely since, for example, the change of a training sample's $A$ would impact the trained model weights $\theta$ and then impact the test fairness on $\theta$. How $A$ would impact the final fairness is complex and therefore we need the influence function and our whole approach of generating counterfactual samples to estimate it.
>
> ### **3. "override(.)" Operator**
>
> We thank the reviewer for the clarification. As the reviewer already noticed our explanation in Section 2.3, we will try our best to further clarify.
>
> The "override(.)" operator is conceptually similar to the do(.) operator in the causal inference; but it is broader. In general, it represents the change of data attributes (feature $X$, label $Y$, or sensitive attribute $A$) of training samples. Since $X$, $Y$, and $Y$ are interdependent, we need to find out, for example, how the change of $A$ would impact $X$. If we already know the underlying causal relationship between variables, then it is the same as the do(.) operator. However, it would only work on synthetic data.
>
> In practice, given non-synthetic data, the causal relationship is unclear (and discovering it is a notoriously hard problem in causal inference), which is our focus where the traditional causal methods are not applicable. To this end, we propose a set of empirical methods to estimate it (i.e. our W-GANs).
>
> ### **4. Baseline Comparison**
> We thank the reviewer for raising the concern.
>
> To the best of our knowledge, the only prior works that study influence function on fairness are [1, 2]. However, both are not comparable to our work because they only consider removing or reweighing whole training samples while we decompose the fairness influence into features, labels, and sensitive attributes. We do not know other works that also study the fairness influence function of modifying features, labels, and sensitive attributes *independently*. If the reviewer knows, please let us know.
>
> In addition, the "Removal" baseline in our experiment considers the effect of removing a training sample, which can be thought of as the method commonly used in the literature [1,2].
>
> Furthermore, we have compared our mitigation to the baseline of in-processing mitigation with fair training [3] in Figure 5.
>
> Last but not least, our work is mainly an explainable AI tool and the primary goal is to explain fairness to AI practitioners. The main objective is not to outperform other bias mitigation methods but rather attributing and explaining the observed fairness.
>
> [1] Wang, Jialu, Xin Eric Wang, and Yang Liu. "Understanding instance-level impact of fairness constraints." International Conference on Machine Learning. PMLR, 2022.
>
> [2] Li, Peizhao, and Hongfu Liu. "Achieving fairness at no utility cost via data reweighing with influence." International Conference on Machine Learning. PMLR, 2022.
>
> [3] Agarwal, Alekh, et al. "A reductions approach to fair classification." International conference on machine learning. PMLR, 2018.

---

> > ### Author Response · Authors · 2023-11-15
> > **Rebuttal Part 2**
> >
> > ### **5. Explicit Definitions on Fairness**
> > We thank the reviewer for clarification. We have included all precise mathematical definitions of fairness in Appendix C, Table 1's middle column. We understand it is easy to omit them. We included them in the Appendix because they are the standard group fairness definitions and we thought no special mention is needed. We hope it also answers Question (2).
> >
> > Therefore, when we say "fairness improves," it simply means the measured unfairness, i.e. the gap between different groups, becomes smaller. They are indeed the standard phrases in the fairness literature, as the reviewer pointed out, and we do not invent anything new. We will clarify it.
> >
> > We hope it is also clear that our fairness definition is mathematical rather than social which is not up to subjective interpretations.
> >
> > ### **6. Limitations**
> > We thank the reviewer for the concern. We did not explain some of the design choices for using HVP in the influence function because it is well-documented in the literature [1]. For example, we use HVP because directly computing and investing Hessian is computationally costly. When the model becomes large, the convergence of HVP would suffer, which is well-documented [2]. We admit our method would be impacted as well. We have admitted them at the end of Section 3.2; we will further clarify them in the paper.
> >
> > However, also note that our goal is not to propose a better influence function approximating algorithm - itself probably warrants being a separate work. Our purpose is to demonstrate the idea of leveraging influence function to help practitioners understand the unfairness. As better influence-approximating algorithms are invented, our framework is flexible enough to plug in and benefit from the improvement in that field.
> >
> > [1] Koh, Pang Wei, and Percy Liang. "Understanding black-box predictions via influence functions." International conference on machine learning. PMLR, 2017.
> >
> > [2] Basu, Samyadeep, Philip Pope, and Soheil Feizi. "Influence functions in deep learning are fragile." Proc. of ICLR, 2021.
> >
> >
> > ### **7. Clarifying A Sentence**
> > "We show that the influence function often identifies the data from the majority group and recommends them to be changed to the minority group." It does not mean reweighting training samples but rather resampling training samples. It means our method mostly recommends counterfactually changing training samples from a disadvantaged group (e.g. female) to an advantaged group (e.g. male), i.e. had those female applicants were to become male applicants, the fairness of the application decisions would improve. It is similar to resampling to increase female representation in the training data to reduce gender bias.

---

> > > ### Author Response · Authors · 2023-11-22
> > > **Discussion period is ending**
> > >
> > > We thank the reviewer for the service. We have responded to the review and clarified a variety of questions. Since the discussion period ends within 24 hours, please let us know if the reviewer still has any unresolved concerns.

---

### Official Review · Reviewer_BzqW · 2023-11-01

**Soundness:** 2 fair
**Presentation:** 2 fair
**Contribution:** 2 fair
**Rating:** 5
**Confidence:** 4

**Summary:**

This paper introduces CIF (concept influence for fairness) as a framework to identify potential fairness problems in training data when looking at 1) sensitive attributes 2) features and 3) labels by overriding concepts in each case. The authors provide details on how to change concepts, and then how to define the influence of those concept changes. The authors provide experimental results.

**Strengths:**

The paper provides a framework that is very flexible to changing technologies.

**Weaknesses:**

This paper seems incremental in its approach. For example, the authors state that the effectiveness of their solution depends on finding a proper transform() which is their work's focus. The then proposed transforms in "generating counterfactual samples" use techniques not novel to this paper and pulled from a number of different sources. Although this framework is generally useful, I don't believe it provides enough novelty for this conference.

**Questions:**

Could the authors provide more details on the "overriding X" experiment w.r.t each of the datasets used. What kind of features where chosen in these cases?

Could the authors please comment on novelty from above?

---

> ### Author Response · Authors · 2023-11-15
> **Rebuttal**
>
> We thank the reviewer for considering our work to be "generally useful." We address the concerns one by one.
>
> ### **1. Incremental Approach**
> We thank the reviewer for the concern. We understand that it might look like our approach is incremental: influence function and W-GAN are adapted and modified from the prior works. However, our novelty does not lie in the details of measuring the influence of samples or generating counterfactual samples; it rather lies in the proposal of a general *framework* that explains and measures the training samples' influence on fairness.
>
> To the best of our knowledge, almost no work in the area of influence function is based on a completely new technical approach. For example, influence function itself [1] is merely a revival of the concept in robust statistics that can be traced back to as early as 1974.
>
> In addition, our framework is far from something technically trivial. None of the prior works can be directly applied in our setting. For example, the original influence function does not target fairness, and we adapt it to the fairness context. Furthermore, W-GAN in [2] only works on sensitive attributes, and we adapt it to also consider features and labels.
>
> Most importantly, the concept of our framework is entirely new. The limited prior works on understanding fairness via influence function [3,4] so far only focus on removing or reweighing whole training samples. Our framework decomposes the fairness influence into features, labels, and sensitive attributes. This distinction is vital in fairness because each attribute can significantly impact the underlying fairness. And it is feasible because of our novel framework by generating counterfactual samples.
>
> In short, our framework is more flexible, powerful, and more suitable for the nature of AI fairness and can give practitioners a wider scope of understanding of fairness. It also leads to a wider range of applications, e.g. detecting mislabeling, fixing imbalanced representations, and detecting fairness-targeted poisoning attacks, as articulated in the paper.
>
> We think if we all insist on an entirely new invention of all methodological details, then almost no work on fairness, influence function, or causal inference can be published on ICLR. We believe most of the papers in those areas do not propose a *completely* new technique.
>
> We will clarify it in the paper. Please let us know if we still have not addressed the concern. If the reviewer disagrees, feel free to let us know the reasons as well as some concrete and technically detailed examples.
>
> [1] Koh, Pang Wei, and Percy Liang. "Understanding black-box predictions via influence functions." International conference on machine learning. PMLR, 2017.
>
> [2] Black, Emily, Samuel Yeom, and Matt Fredrikson. "Fliptest: fairness testing via optimal transport." Proceedings of the 2020 Conference on Fairness, Accountability, and Transparency. 2020.
>
> [3] Wang, Jialu, Xin Eric Wang, and Yang Liu. "Understanding instance-level impact of fairness constraints." International Conference on Machine Learning. PMLR, 2022.
>
> [4] Li, Peizhao, and Hongfu Liu. "Achieving fairness at no utility cost via data reweighing with influence." International Conference on Machine Learning. PMLR, 2022.
>
>
> ### **2. Details of "Overriding X" Experiments**
> We thank the reviewer for asking for clarification. Due to the space limit, we have included the settings of overriding $X$ in our experiments in Appendix E.1. We understand that it is easy to be omitted. We recap it here.
>
> "Overriding X" means counterfactually changing samples as if their features were changed to different values, and then measuring how the underlying fairness would be impacted. For example, in an image task, if we want to understand how skin color would affect fairness (say gender fairness), we can generate counterfactual features (image pixels) as if they come from a different race, i.e. how would fairness change if those images change their racial group.
>
> In COMPAS, $X$ is the binary feature that indicates age $> 45$ or not while the sensitive attribute is race. In Adult, $X$ is race while $A$ is sex. In CelebA, $X$ is the binary label ``Young'' (i.e. age) while $A$ is gender.
>
> Please let us know if this is still not clear enough.

---

> > ### Author Response · Authors · 2023-11-22
> > **Discussion period is ending soon**
> >
> > We thank the reviewer for the service. We have already responded to the reviewer's concern about technical contribution and clarified the experimental setting in the rebuttal. Since the discussion period will end within 24 hours, please let us know if the reviewer still has any questions.

---

### Author Response · Authors · 2023-11-15
**Global Response**

We thank the positive feedback from the reviewers, i.e. considering our work to be "generally useful" (Reviewer BzqW), "important" (Reviewer rxAc), "well-written," (Reviewer AAoD) "intriguing," (Reviewer AAoD) "practical," (Reviewer AAoD
) "intuitive," (Reviewer AAoD) "easy to understand", (Reviewer AAoD) and "reasonable design." (Reviewer AAoD)

The majority of the reviews are clarifications and we believe we have successfully addressed them. For example, the contribution (Reviewer BzqW), applicability (Reviewer rxAc), the relationship between concepts and fairnesses (Reviewer rxAc), override(.) operator (Reviewer rxAc), baseline comparison (Reviewer rxAc), fairness definition (Reviewer rxAc), and theoretical understanding (Reviewer AAoD).

We believe there are no technical concerns about our work that remain unsolved at this point. Please let us know if there is still anything unclear.

Thanks.

Authors of Paper 3936

---

### Meta-Review · Area_Chair_sPqp · 2023-12-09

**Metareview:**

This paper introduces CIF from the influence function literature to quantify algorithmic fairness in training data. The authors investigate 3 cases including sensitive attributes, features and labels by overriding concepts.

This paper provides a flexible framework on an important topic (fairness) in machine learning. However, there are concerns on the novelty of the method and concerns on the writing, which I also agree after reading it. A few important parts are deferred to the appendix or missing (proofs, metric details, definition, baselines), and the clarity can be improved for technical parts as also mentioned by the reviewers. It is encouraged that the authors to consider reorganizing the papers to incorporate reviewers suggestions.

Based on above reasons, I recommend rejection for this paper.

**Justification For Why Not Higher Score:**

the quality of the draft can be improved

**Justification For Why Not Lower Score:**

N/A

---

### Decision · Program_Chairs · 2024-01-16

Reject